# Understanding the Polymerization of Polyfurfuryl Alcohol: Ring Opening and Diels-Alder Reactions

**DOI:** 10.3390/polym11122126

**Published:** 2019-12-17

**Authors:** Gianluca Tondi, Nicola Cefarin, Thomas Sepperer, Francesco D’Amico, Raphael J. F. Berger, Maurizio Musso, Giovanni Birarda, Andreas Reyer, Thomas Schnabel, Lisa Vaccari

**Affiliations:** 1Forest Products Technology & Timber Constructions Department, Salzburg University of Applied Sciences, Marktstrasse 136a, 5431 Kuchl, Austria; thomas.sepperer@fh-salzburg.ac.at (T.S.); thomas.schnabel@fh-salzburg.ac.at (T.S.); 2Salzburg Center for Smart Materials, Jakob-Haringer-strasse 2, 5020 Salzburg, Austria; 3Land, Environment, Agriculture and Forestry Department, University of Padua, Via dell’Università 16, 35020 Legnaro (PD), Italy; 4Elettra-Sincrotrone Trieste S.C.p.A., Strada Statale 14—km 163,5 in AREA Science Park, 34149 Basovizza, Trieste, Italy; nicola.cefarin@elettra.eu (N.C.); francesco.damico@elettra.eu (F.D.); giovanni.birarda@elettra.eu (G.B.); lisa.vaccari@elettra.eu (L.V.); 5Department of Chemistry and Physics of Materials, Paris Lodron University Salzburg, Jakob-Haringer-Strasse 2a, 5020 Salzburg, Austria; raphael.berger@sbg.ac.at (R.J.F.B.); maurizio.musso@sbg.ac.at (M.M.); andreas.reyer@sbg.ac.at (A.R.)

**Keywords:** furanic polymer, spectroscopy, solid-state ^13^C-NMR, FTIR, Raman, linear structure, ring opening, γ-lactone, Diels-Alder

## Abstract

Polyfurfuryl alcohol (PFA) is one of the most intriguing polymers because, despite its easy polymerization in acid environment, its molecular structure is definitely not obvious. Many studies have been performed in recent decades, and every time, surprising aspects came out. With the present study, we aim to take advantage of all of the findings of previous investigations and exploit them for the interpretation of the completely cured PFA spectra registered with three of the most powerful techniques for the characterization of solid, insoluble polymers: Solid-State ^13^C-NMR, Attenuated Total Reflectance (ATR), Fourier Transform Infrared (FTIR) spectroscopy, and UV-resonant Raman spectroscopy at different excitation wavelengths, using both an UV laser source and UV synchrotron radiation. In addition, the foreseen structures were modeled and the corresponding ^13^C-NMR and FTIR spectra were simulated with first-principles and semi-empiric methods to evaluate their matching with experimental ones. Thanks to this multi-technique approach, based on complementary analytical tools and computational support, it was possible to conclude that, in addition to the major linear unconjugated polymerization, the PFA structure consists of Diels-Alder rearrangements occurring after the opening of some furanic units, while the terminal moieties of the chain involves γ-lactone arrangements. The occurrence of head-head methylene ether bridges and free hydroxyl groups (from unreacted furfuryl alcohol, FA, or terminal chains) could be excluded, while the conjugated systems could be considered rather limited.

## 1. Introduction

The polymerization of furfuryl alcohol (FA) in acid environment to polyfurfuryl alcohol (PFA) is a subject that has attracted the interest of many scientists between the 1950s and today. This dark thermosetting polymer has been investigated in consideration of various synthetic and analytic perspectives, and new information about its structure has always jumped out. In Table 1, some of the most important studies on PFA are summarized by presenting the molecular arrangements proposed by the authors.

The first investigation of this intriguing polymer was published by Dunlop and Peters in 1953 [1], presenting the linear furanic polymerization through methylene (**1**) and di-methylene ether bridges. This model explained the linear structures and attributed the crosslinking to other methylene bridges obtained from the formaldehyde (coming from the head-head condensation of two molecules of FA). During the early 1960s, Conley and Metil [2] performed their studies by means of FTIR spectroscopy, where they identified the ring opening due to the formation of γ-diketones (**2**). A decade later, Wewerka [3] proposed a polymerization pathway for the termination of the chains through formation of a γ-lactone (**3**), which occurred for its alumina-catalyzed PFAs. Specifically, in the aforementioned study, a large amount of low polymerized units were produced and an increased amount of γ-lactone was observed. In 1984 Chuang et al. [4] first published ^13^C-NMR results on PFA, stating that the presence of dimethylene ether bridges was very limited, as well as the substitution of C3 and C4 of the furanic ring. The Chuang group observed a peak above 200 ppm, assigned to the presence of γ-diketones due to ring opening and confirming the observation of Conley and Metil [2], and also a small signal at 177 ppm that they attributed to levulinic acid. This study supported crosslinking through formaldehyde-derived methylene bridges (**4**) first proposed by Dunlop and Peters [1]. One year later, Buchwalter [5] published the ^1^H and ^13^C-NMR analysis on furfuryl acetate, and it was the first study to propose a conjugated structure (**5**). Hence, in 1996, Choura et al. [6,7] published a systematic study on the polymerization of FA using several model compounds [6]. In this study, which exploited FTIR spectroscopy and ^1^H-NMR as analytic tools for the characterization of the polymers, the research group concluded that the polymerization of FA occurs in two steps: the first was linear polymerization though methylene bridges (as already proposed by Dunlop and Peters [1]), while the second was far more complex than classically expected. In particular, the chromophore effect of the polymer was attributed to the conjugation of the linear furanic chain, while the branching was expected to involve Diels-Alder reactions between conjugated and unconjugated chains (**6**) [7]. These solid studies did not involve ^13^C-NMR investigations, and therefore did not completely consider the possibility of ring opening. In the 2000s, other scientists came across this polymer: Bertarione et al. [8] focused on the polymerization of FA in confined spaces and studied this macromolecule with several vibrational techniques, suggesting the polymerization mechanism to occur through carbocations. Guigo et al. [9,10] studied the chemo-rheological kinetics of this polymerization, confirming again the two reaction steps and attributing the curing step to Diels-Alder reactions. Barsberg and Thygesen [11] studied PFA formation using Attenuated Total Reflectance (ATR) FTIR spectroscopy, supporting their spectral attributions with theoretical predictions using density functional theory of the vibrational bands of the most probable PFA model structures. The aforementioned study confirmed the dominance of the methylene bridge formation in the initial polymerization phase and the formation of conjugated systems in the second one, supporting the model of Buchwalter. PFA curing was also studied by Raman spectroscopy, and in this field Kim et al. [12,13] analyzed the acid polymerization of FA focusing on the thermodynamics of the system and suggesting that, based on the comparison of measured and calculated Raman spectra, the formation of a conjugated diene structures (the conjugation of the furanic group) was preferred to the diketonic structures.

Recently, Falco et al. [14] presented a study in which major importance was given to the opening of the furanic ring, confirming the presence of Diels-Alder crosslinking. Both the study of the group of Gandini [6,7] and that of Falco agreed on the presence of Diels-Alder arrangements, but Falco et al. [14] gave major importance to the ring opening and also confirmed the inter-chained methylene bridge which was “almost” excluded by Gandini.

Overall, we can conclude that there is no general consensus on the possible structure(s) of PFA, and more comprehensive studies are needed to further clarify the structure of the reaction products. To shed more light on the PFA molecular structure after polymerization completion, we performed an analysis of PFA synthesized through simple acid catalysis at room temperature by complementing spectroscopic techniques, such as solid-state ^13^C-NMR, ATR FTIR and Raman spectroscopy with UV laser excitation and using UV synchrotron light, with theoretical modeling.

## 2. Materials and Methods

### 2.1. Sample Preparation

PFA was prepared at room temperature by adding 0.1 mL of sulfuric acid (Merck) 32% in an open test tube containing 2 mL FA (Transfuran chemicals, Geel, Belgium). The reaction started with a darkening of the original yellow solution, which developed into an orange, then brown solution that turned into a black solid after around 60–80 s through exothermal reaction. The solid polymer was kept in an oven at 103 °C for 1 h to ensure the complete curing and then it was grinded into fine powder. This powder was leached with water to remove the catalyst and the unreacted FA, and then dried again at 103 °C for 4 h until constant weight. The material resulted almost completely insoluble (only 2% was leached out) and the resulting dried powder was the subject of this study.

### 2.2. Solid-State ^13^C-NMR

The ^13^C-NMR spectrum of PFA powder was obtained with a Bruker Avance NEO 500 wide bore system (Bruker BioSpin, Rheinstetten, Germany) at the NMR centre of the Faculty of Chemistry at the University of Vienna. A 4 mm triple resonance magic angle spinning (MAS) probe was used with a resonance frequency for ^13^C of 125.78 MHz, and the MAS rotor spinning was set to 14 kHz. Cross polarization (CP) was achieved by a ramped contact pulse with a contact time of 2 ms. During acquisition, ^1^H was high-power decoupled using SPINAL with 64 phase permutations. The chemical shifts for ^13^C are reported in ppm and are externally referenced to adamantane by setting the low field signal to 38.48 ppm.

The data elaboration was done with the software Top-spin 4.0.6 (Bruker) and OriginPro (OriginLab, Northampton, MA, USA), while the calculations of the theoretical chemical shifts were done with the software NMR-Predict developed by the University of Lausanne (L. Patiny) and the University of del Valle (J. Wist) [15,16,17].

### 2.3. ATR-FTIR Analysis

The FTIR experiments were carried out at the Chemical and Life Sciences branch of the Synchrotron Infrared Source for Spectroscopy and Imaging beamline (SISSI-Bio) at the Elettra Sincrotrone Trieste, Italy [18].

PFA powder was measured by ATR-FTIR spectroscopy using the Platinum™ Single Reflection ATR box (Bruker Optik GmbH, Ettlingen, Germany) with diamond internal reflective element (IRE). Spectra were acquired using the Vertex 70 v in-vacuum interferometer (Bruker Optik GmbH, Ettlingen, Germany) equipped with a wide band deuterated triglycine sulfate detector and silicon FIR-MIR beamsplitter. ATR-FTIR spectra were collected from 6000 to 60 cm^−1^ in double side, forward/backward acquisition mode with a scanner velocity of 5 kHz. For each spectrum, 256 and 128 scans were averaged with a spectral resolution of 2 cm^−1^ for the background and the sample respectively. Fourier transform was carried out with Mertz Blackman-Harris 3-terms apodization function. Each sample was measured 5 times and the results averaged before comparison and further analyses. Background was collected on clean diamond IRE. The spectra were cut in the 4000 to 400 cm^−1^ region, then baseline corrected with the concave rubberband function (5 iterations) and finally vector normalized (4000 to 400 cm^−1^ region) and corrected for the offset with OPUS 7.5 (Bruker Optik GmbH, Ettlingen, Germany) software. Peak assignment procedure was guided by second derivative analysis (Savitzky-Golay algorithm, 19-smoothing points) in order to enhance the separation of overlapping peaks.

### 2.4. Raman Investigation

UV Resonant Raman (UVRR) measurements were carried out at the IUVS beamline of the Elettra Sincrotrone Trieste, Italy [19]. The synchrotron-based radiation source was employed to get the excitation wavelengths of 226 nm and 249 nm, with a beam power reaching the sample of approximately 10 μW. A UV laser source has been adopted for the excitation wavelength at 266 nm with 100 µW beam power. The Raman signal was collected in backscattering configuration. During the measurements, the samples were continuously moved with oscillations of 1 Hz and 1 mm to avoid photodegradation phenomena. A Czery-Turner spectrometer with focal length of 750 mm (Acton SP-2750, Acton, MA, USA), coupled with a holographic reflection grating of 1800 g/mm and with a Peltier-cooled back-thinned CCD (Princeton Instruments 7510, Acton, MA, USA), was employed to collect the Raman signal. Spectral resolution (FWHM of the apparatus function) was set to 25 cm^−1^ for the synchrotron-based Raman measurements and to 7 cm^−1^ for the UV laser-based Raman measurements. Raman wavenumbers were calibrated by using cyclohexane spectra [20].

### 2.5. Computational Details

The starting geometries for the model structures (**1**), (**2**), (**3**) and (**7**) (see Table 1) were set up using Marvin suite [21], and the structures were pre-optimized with Marvin’s molecular dynamics module using standard settings. The resulting lowest energy conformer was used in all four cases for the further steps. The pre-optimized structures were optimized with Turbomole [22] at the ri-DFT level of theory [23] using the BP86 functional [24,25] together with the RI and MARIJ [23] approximations, the def-SV(P) basis set [26,27] and standard settings otherwise. After geometry convergence, a single point DFT calculation with tightened numerical convergence criteria ($denconv 0.1d–07 and $scfconf 8) prior to the NMR shielding calculation with Turbomoles mpshift module was performed [28]. Relative isotropic chemical shifts with respect to tetramethylsilane (TMS) were calculated (the computation of TMS and its properties was performed in a fully analogous way) for all carbon atoms for the selected (**1**), (**2**), (**3**) and (**7**) molecules. Intensities of 0.6 for (**1**), 0.1 for (**2**), 0.1 for (**3**), and 0.2 for (**7**), respectively, were assigned to the signals (irrespective of the chemical connectivity of the carbon atoms) to obtain rough ^13^C-NMR model spectra. All four model spectra were merged, and the merged spectrum was folded with a Lorentzian line shape of 5 ppm half-width. Finally, the chemical shielding values were scaled by a factor of 108/120 in order to obtain a match of the respective peak at 108 ppm in the experimental ^13^C-NMR spectrum.

For the calculation of the IR model spectrum, the lowest energy conformer search for all four structures from the DFT optimization has been performed with Grimme’s xtb method and the “confscript” script using all standard settings [29]. For the obtained minimum conformers, a hessian calculation using the hessian module and standard settings of xtb was performed to arrive at IR frequencies and intensities. To obtain the model spectrum, as in the case of the NMR spectra, the four individually calculated IR spectra were scaled and merged (0.6 (**1**) + 0.1 (**2**) + 0.1 (**3**) + 0.2 (**7**)), and folded in this case with a Gaussian line shape of 0.05% of the full width of the spectrum (20 cm^−1^) as half width.

## 3. Results and Discussion

The numerous studies performed on PFA are fully justified, because this polymer cannot be explained by simple chemistry. Almost all authors agreed that this polymerization does not follow a single pattern, but many arrangements may occur simultaneously, and they also depend on the procedure adopted. The approach we consider in the present paper consists of evaluating the proposed arrangements summarized in Table 1, discriminating between those structures that are possible and those that are not on the basis of the results obtained by ^13^C-NMR, FTIR and Raman spectroscopies on our thermoset PFA sample, and finally comparing the registered spectra with the computational data to strengthen the spectral interpretation.

### 3.1. Solid-State ^13^C-NMR

The solid-state ^13^C-NMR spectrum of PFA is reported in Figure 1.

In Table 2, the interpretation of the ^13^C-NMR signals is summarized, and the presence of the identified chemical moieties in the possible PFA structures of Table 1 is highlighted.

The ^13^C-NMR results, summarized in Table 2, suggest that all the presented structures contribute to explaining the signals. However, the major contribution to the PFA structure is certainly given by the linear structure (**1**). Indeed, the high intensities of the signals at 151, 108 and 27 ppm suggest that this structure represents a major proportion of the whole macromolecule, even if it alone cannot justify the presence of the other signals. The broad peak at 38 ppm can be attributed to Diels-Alder arrangements (**6**,**7**) and to the methylene bridge structure (**4**) only, while the presence of a shoulder at around 49 ppm can be attributed to the methylene bridge between furans and carbonyl groups (**2**,**7**). The latter are certainly present due to the signals of the keto-carbonyl at 219 and 204 ppm. The signal at 142 ppm is due to CH=CH inside the unsaturated ring, while the signal at around 88 ppm is due to quaternary carbons. Both assignments can be attributed to Diels-Alder arrangements (**6**,**7**) and also to γ-lactone formation (**3**). The presence of the bands at 13 and 172 ppm can be attributed exclusively to the methyl and carbonyl groups of the lactone (**3**), which possibly occurs in the terminal units of the chain.

The relative intensities of these signals suggest that ring opening (**2**,**7**) and Diels-Alder (**6**,**7**) are secondary structural arrangements of completely cured PFA, while γ-lactone (**3**) occurs in a lower percentage. In addition, the assignments given in Table 2 also make it possible to postulate the presence of conjugated structures (**5**, **6**), since the region at around 126 and 94 ppm, presenting some broad peaks, cannot be completely justified by the presence of γ-lactone structures (**3**) and ring opening (**2**). Nevertheless, the more probable Diels-Alder occurrence pathway should be the one presented in structure (**7**), with the γ-diketonic structure (**2**) being more abundant and more dienophilic than conjugated furanic (**5**). We propose a mechanism that encompasses the ring opening of some furanic rings, producing a double bond which promptly combines with the linear PFA to produce structure (**7**) through Diels-Alder crosslinking (See Figure 2).

Moreover, the presence of structure (**4**) could not be excluded, since the broad signal at 38 ppm might be due to both -CH- methyne bridged furanic rings in structure (**4**) and tertiary carbons in Diels-Alder products (**6**,**7**), while, at the same time, the signal at 27 ppm of -CH_2_- methylene bridge overlaps with the methylene bridge of structure (**1**). In this investigation, the presence of levulinic derivatives, proposed by Chuang et al. [4], is less probable because simple acid and esters will be leached out.

In summary, on the basis of the observations of the solid-state ^13^C-NMR spectrum of hardened PFA, it is possible to assert that the dominant PFA structure is the linear one (**1**), while structures (**2**) and (**7**) significantly contribute to the final composition of the polymer, and that structure (**3**) may occur in the terminal part of the chains. Conjugated systems (**5**,**6**) and the methylene bridged structure (**4**) could not be clearly identified and anyway their contribution to the final structure should be negligible with respect to the other proposed arrangements. At the same time, solid-state ^13^C-NMR excludes the presence of: (i) dimethylene-ether bridges (head-head combination of FA molecules [1]); (ii) unreacted FA; and (iii) opening of two consecutive furanic rings, since β-diketones signals were not detected.

### 3.2. ATR-FTIR Spectroscopy

The ATR-FTIR spectrum of PFA registered with conventional Mid-IR sources is reported in Figure 3, while the attributions of most of the diagnostic peaks identified here are summarized in Table 3, following the same organization scheme proposed in Table 2.

As a preliminary observation from the analysis of Figure 3, it is evident that the broad band that characterizes the FA in the 3600–3000 cm^−1^ spectral region (see Appendix A for FA spectrum), associated with hydroxyl group stretching, is not present in the hardened PFA spectrum, confirming that the percentage of unreacted FA is below the detection limit of the technique. At the same time, for the same reason, it is possible to exclude the presence of levulinic acid.

Analyzing the proposed structures, we can observe that the signals expected for structure (**1**) are all present and give major absorptions at 1560, 1420 and 780 cm^−1^, which is in agreement with the ^13^C-NMR findings. Structure (**2**) is also certainly present, as proven by the signals of the α,β-unsaturated ketone at around 1690 and 1670 cm^−1^_,_ assigned to the carbonyl group conjugated with the vicinal C=C, and at 1615–1590 cm^−1^, related to the carbon-carbon double bond. The presence of structure (**3**) is also possible. Indeed, the α,β-unsaturated γ-lactones generate two intense and characteristic peaks in the range between 1790–1740 cm^−1^, due to the stretching of the carbonyl group, and two more components in the 1345–1290 cm^−1^ spectral window, usually assigned to the C-O stretching of the lactone ring [30]. Although the aforementioned signals are present, their intensities are quite low. Therefore, structure (**3**) could be present only in small amounts, confirming this pattern as a possible end-chain arrangement. Structure (**4**) with methylene bridges can also not be excluded, because the aliphatic stretching region, between 3000–2800 cm^−1^, is composed of several subcomponents, which may be explained by C-H stretching modes of different moieties, including the methyne and methylene bridges. The signals possibly generated by the conjugated structure (**5**) have already been discussed in previous papers [11,12,31], but an unambiguous assignment of the vibrational modes to the peaks in the infrared spectra has still not yet been established, even if vibrational analysis has been supported by theoretical simulations. As a matter of fact, the peaks at 1650 cm^−1^ and 1600 cm^−1^ could be assigned either to diene structures [31], or as reported in Table 3. Therefore, on the basis of the literature, FTIR alone cannot exclude structure (**5**), and hence (**6**). At the same time, structures (**2**) and (**7**) explain all the remaining peaks in the 1710−1520 cm^−1^ region. Specifically, the presence of the Diels-Alder rearrangement product (**7**) seems to be very reasonable, due to the presence of two distinctive signals: the isolated C=O stretching at 1710 cm^−1^ and the isolated C=C stretching at 1650 cm^−1^; formed after the reaction between the furan ring (diene) and the carbon-carbon double bond of the structure (**2**) (dienophile), accordingly with the reactions scheme proposed in Figure 2.

Finally, it is also possible to point out that all the signals present in the 1710–1520 cm^−1^ spectral window, mostly involving C=O and C=C vibrational modes, are directly correlated with the polymerization reaction. Indeed, no signals are present for the FA spectrum in the aforementioned region (see Appendix A), since they are due to the new chain formations and rearrangements. This highlights that the ring opening (**2**) certainly occurs and that lactone arrangement (**3**) is reasonable.

### 3.3. UV-Raman Spectroscopy

To verify the presence of conjugation inside the PFA sample, we performed Raman measurements employing excitation wavelengths of 266 nm, 259 nm and 226 nm. The collected spectra are plotted in Figure 4. At the chosen excitation wavelengths, resonance conditions with the aromatic compounds and the conjugated part of the polymer are achieved. As a matter of fact, as reported by Asher et al. [32,33], UV-Raman spectroscopy uses selective excitation in the UV absorption bands of molecules to produce spectra of their chromophoric segments, enhancing the selectivity of the technique. Specifically, conjugated π-bond systems act as chromophores, exhibiting an absorption wavelength the longer the higher the conjugation degree. In addition, UV-Raman measurements of condensed-phase samples excited below 270 nm are negligibly plagued by fluorescent interferences, further improving the spectral quality.

The careful comparison of our spectra with those obtained by Kim et al. [12,31] evidences important differences, which can be addressed by a different polymer composition, possibly due to the curing process. Specifically, the spectra in Kim et al. are characterized by well-defined vibrational peaks with spectral widths lower than 50 cm^−1^ in the 1400–1700 cm^−1^ spectral region. In particular, the Kim spectra are dominated by the vibrational mode centered at 1650 cm^−1^, which has been assigned to conjugated C=C, and therefore to a conjugated-type PFA polymer. On the contrary, our spectra show more broadened peaks, not so well defined as those of Kim et al. [12,31], which can therefore be associated with a complex mixture of different PFA structures. Particularly relevant is the absence of a well-defined, isolated intense peak at 1650 cm^−1^ at all the UV excitation wavelengths employed. Even if, due to the peak broadening, the presence of C=C conjugated vibrational components could not be excluded, at the same time, we can safely assert that the contribution of conjugated structure to the final hardened PFA composition is negligible.

### 3.4. Computational Results

To strengthen the conclusions drawn out complementing the spectroscopic results, a computational effort was made in order to simulate both ^13^C-NMR and FTIR spectra of a mixture of the most probable FPA arrangements, identifiable in the structures (**1**), (**2**), (**3**) and (**7**) of Table 1. It has to be clearly stated that, due to the chemical complexity of the hardened PFA, as is certainly deducible from the mosaic information obtained with ^13^C-NMR, FTIR and UV-Raman analyses, it is almost impossible to precisely model the chemical arrangements of this macromolecule and their relative proportions. For complex mixtures of partially known constituents that result in very broad multi-component FTIR and ^13^C-NMR peaks, only speculative evaluations can be made. On these premises, the final aim of the performed simulations is to demonstrate that our proposed chemical model does not contrast the experimental results. Therefore, for the calculation of the simulated spectra, we tentatively considered a model of the polymer constituted as follows: 60% linear (**1**), 20% Diels-Alder (**7**), 10% ring opening (**2**) and 10% γ-lactone terminal moieties (**3**). In Figure 5, the ^13^C-NMR spectrum calculated on the base of the proposed model (0.6 (**1**) + 0.1 (**2**) + 0.1 (**3**) + 0.2 (**7**)) is compared with the experimental ^13^C-NMR one.

It can be appreciated that the simulated ^13^C-NMR spectrum of the model PFA structure is in good agreement with the experimental one. In particular, the prominent furan-carbon signals at 108 and 151 ppm, as well as the methylene carbon signal at 27 ppm and the carbonyl carbon signals at 204 and 219 ppm, find their close counterparts in the simulated spectra. The greatest gap between the calculated and experimental ^13^C-NMR is due to the signal at 60 ppm (marked by an asterisk) in the simulated spectrum, which has no counterpart in the experimental spectrum. This signal most likely originates from the two terminal furan-units of structure (**7**), the contribution of which is an artifact of modeling the polymer by considering only a short moiety.

Similarly to the ^13^C-NMR spectrum, the simulated IR spectrum also does not contradict the experimental FTIR one (See Figure 6).

In brief, all peaks in the experimental FTIR spectrum also appear in the simulated one. The largest discrepancy between the spectra lies in the two peaks of the simulated spectrum centered at 1350 and 1450 cm^−1^ (marked by asterisks in Figure 6), which do not have a counterpart in the experimental spectrum. The two signals can be assigned to the C=C stretching mode of the two terminal furan moieties in structure (**1**), and their prominent intensity is due to their overweight with respect to the repetitive core, which is also an artifact of the modeling.

To summarize the outcome from the spectroscopic techniques exploited in this work, it is possible to state that structure (**1**), as reported in literature, is the predominant species in this complex matrix. This statement is confirmed by both the ^13^C-NMR and FTIR data. At the same time, the linear aliphatic structure (**1**) might undergo ring opening, and therefore form the γ-diketonic structure (**2**). Both ^13^C-NMR and FTIR suggest that structure (**2**) can still be present in the final PFA product, even if at much lower concentrations with respect to (**1**). Indeed, structure (**2**) likely reacts with structure (**1**) through the Diels-Alder rearrangement, leading to the formation of structure (**7**). Also, this species is highly probable, due to the presence of its characteristic peaks in both ^13^C-NMR and FTIR spectra. The possibility to find conjugated structure (**5**) in this polymer is matter of controversial interpretations. Both the FTIR and UV-Raman were unable to exclude it, since some signals due to this monomeric unit are present. At the same time, these signals might be easily assigned to other species (see Table 3) characterized by the presence of different C=C bonds, as in the case of structures (**2**), (**3**) and (**7**). From the ^13^C-NMR analysis, one broad signal (94 ppm) with low intensity seems to be due to the conjugated moiety. In any case, even if present, the concentration of the conjugated arrangement is negligible with respect to structures (**1**), (**2**) and (**7**), and its eventual Diels-Alder product, (**6**), is less probable than (**7**). Structure (**3**) has some characteristic peaks in the FTIR region due to the α,β-unsaturated γ-lactones, as well as one peak (173 ppm) in the ^13^C-NMR spectrum. The intensities of the aforementioned signals are low, suggesting only a minor presence of this species, possibly as an end-chain of the polymeric mixture. Structure (**4**), characterized by the methylene bridge, is an intriguing polymerization product but its presence is very difficult both to confirm and to deny. Indeed, this structure does not generate any peculiar signal in any of the spectroscopic techniques used in this work so far. The aliphatic moieties (-CH and -CH_2_) are very common and highly repeated functional groups all along the polymer, preventing saying anything about the presence or not of this chemical unit as a product.

Moreover, it is possible to assert that no presence of unreacted FA could be detected, while the possibility of having carboxylic acids in our hardened PFA polymer is also excluded, since there are no signals in the 3600–3300 cm^−1^ region of the FTIR spectra.

Further confirmation of the interpretation reported by the authors comes from the simulated ^13^C-NMR and FTIR spectra. The good matching between the theoretical outcome and the experimental data supports the chemical and structural interpretation of the polymeric PFA mixture just summarized.

## 4. Conclusions

Although this paper focuses its attention on a fairly simple polymerization reaction of a monomeric unit (FA) in acid environment, the final product is not as straightforward as expected from both a chemical and a structural point of view. As highlighted by other authors [9], the polymerization conditions (such as temperature and acid amount) lead to different curing degrees and this affects the relative proportions of the structures proposed in this article. The possible rearrangements and reactions between FA monomers that might occur lead to a wide range of different products, making it possible to postulate that even minimal variations in the synthesis parameters can affect the final PFA composition. The complexity of the final product requires a multi-technique approach that encompasses both experimental and theoretical methods.

By complementing ^13^C-NMR, FTIR and UV-Raman spectroscopies with spectra modelling, it was possible to confirm the complexity of the heterogeneous final structure of the PFA polymer. The interpretation of the data so far collected helped the authors to clarify the presence or not of some structures and to propose a new one as a possible product. In particular, the linear aliphatic polymer (**1**) is the dominant specie in our hardened PFA. At the same time, due to the acidic environment, this structure might evolve into the γ-diketonic structure (**2**) after ring opening. The mutual presence of these two products (**1**,**2**) suggests their Diels-Alder rearrangement in structure (**7**), which has never been proposed before. The presence of structure (**7**) is supported and confirmed by both ^13^C-NMR and FTIR techniques. At the same time, the presence in low concentration of the α,β-unsaturated γ-lactone structure (**3**) seems to be probable as the terminal unit of the polymeric chains.

Despite that in this work three of the most powerful techniques (^13^C-NMR, FTIR, UV-Raman) for investigating solid organic materials were exploited, it was still not possible to achieve an unambiguous overall picture of the FA polymerization reaction. In particular, with respect to the presence of structure (**4**) with methylene bridge, this was almost impossible to assert due to the lack of specific signals. Likewise, from the ^13^C-NMR data, the conjugated structure (**5**) and the Diels-Alder outcome (**6**) seems not to be so favorable, and the same holds true for FTIR and UV-Raman, although they could not exclude their presence. This final remark further strengthens the need for multi-technique approaches for the characterization of macromolecules, even in the case of well-known ones, such as PFA, the complexity of which should never be underestimated.

## Figures and Tables

**Figure 1 polymers-11-02126-f001:**
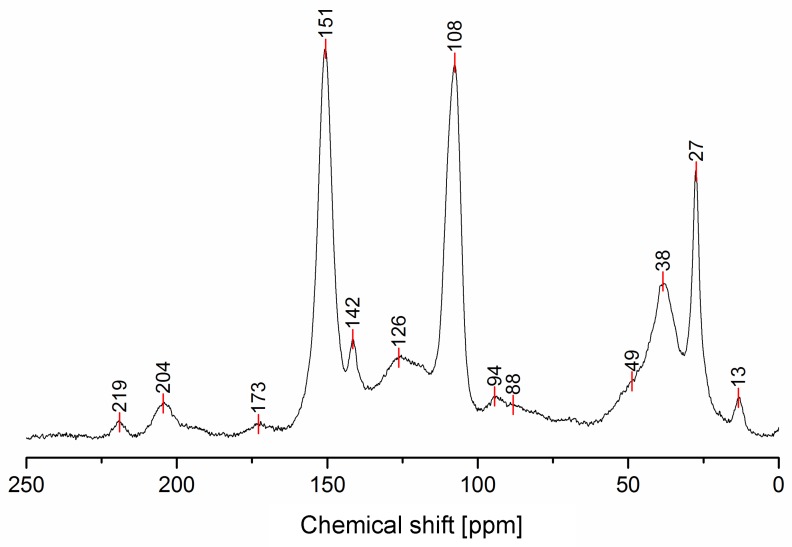
Solid-state ^13^C-NMR spectrum of hardened PFA.

**Figure 2 polymers-11-02126-f002:**
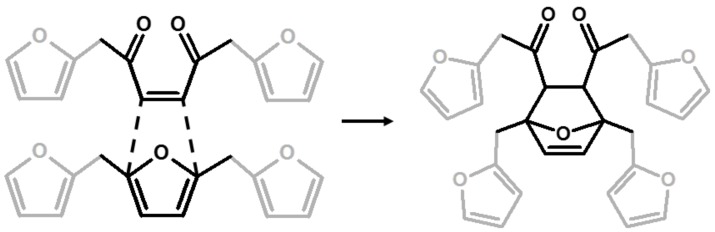
Diels-Alder crosslinking reaction between linear (**1**) and ring opened (**2**) PFA structures.

**Figure 3 polymers-11-02126-f003:**
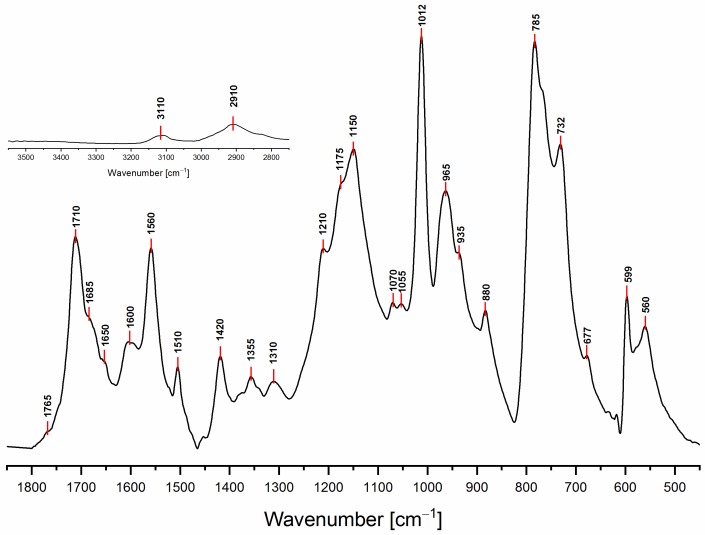
FTIR spectrum of hardened PFA in the spectral region 1850–450 cm^−1^. In the inset, the spectral region 3550–2750 cm^−1^ is plotted, keeping the original intensity scale.

**Figure 4 polymers-11-02126-f004:**
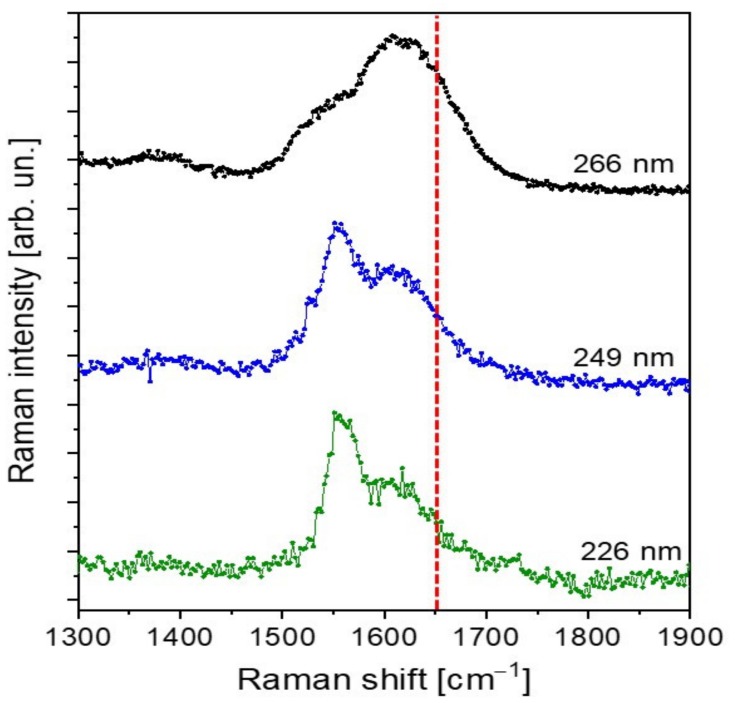
UV-Raman spectra of hardened PFA collected with UV laser excitation at 266 nm (black curve), and with synchrotron radiation excitation at 249 nm and 226 nm (blue and green curves respectively). Red dotted line pinpoints the 1650 cm^−1^ wavenumber.

**Figure 5 polymers-11-02126-f005:**
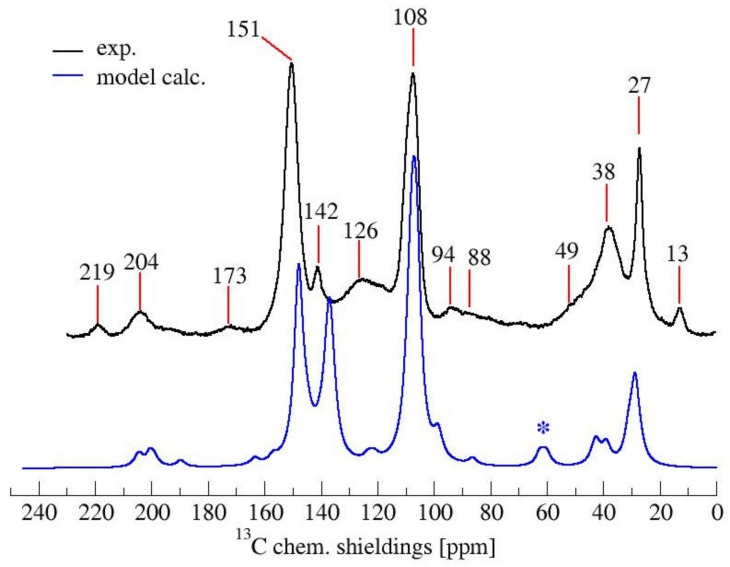
Comparison of experimental and calculated ^13^C-NMR spectrum of PFA. With * the bands due to the artifact of the model.

**Figure 6 polymers-11-02126-f006:**
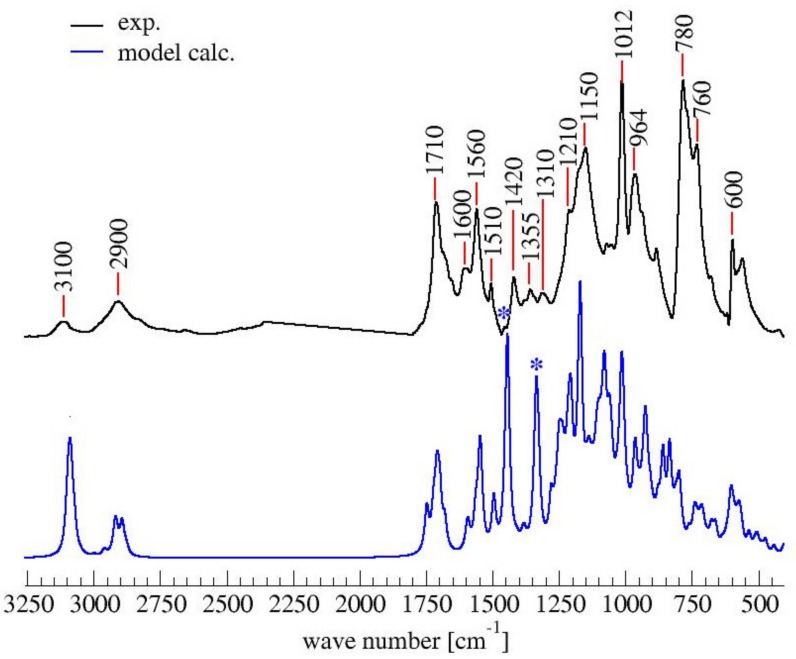
Comparison of experimental and calculated FTIR spectrum of PFA. With * the bands due to the artifact of the model.

**Table 1 polymers-11-02126-t001:** Possible structural arrangements of PFA, their chemical formula and the related scientific papers that first proposed them. Repetitive PFA core moieties are drawn in black, while furanic rings in grey represent the terminal units.

Structure	Arrangement	Chemical Formula	Reference(s)
1	Linear	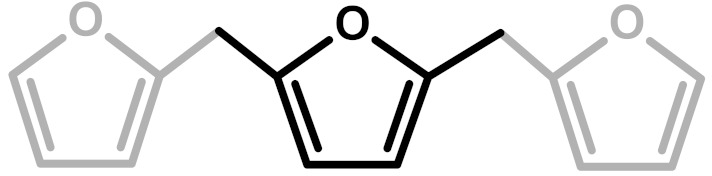	Dunlop & Peters—1953 [1]
2	Ring opening	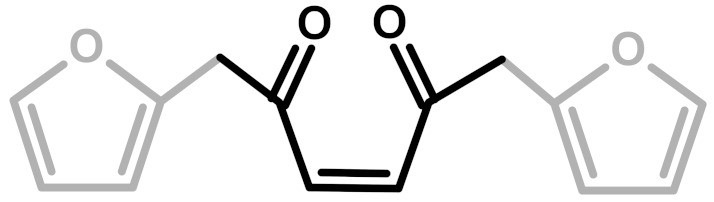	Conley & Metil—1963 [2]
3	α,β-unsaturated γ-lactons	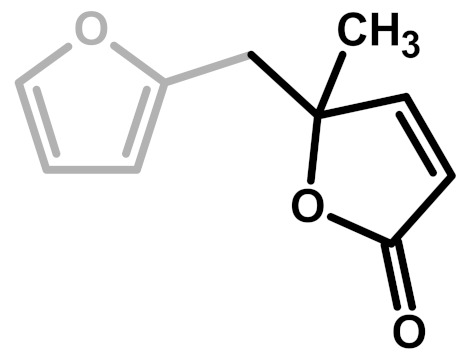	Wewerka—1971 [3]
4	Methylene bridge	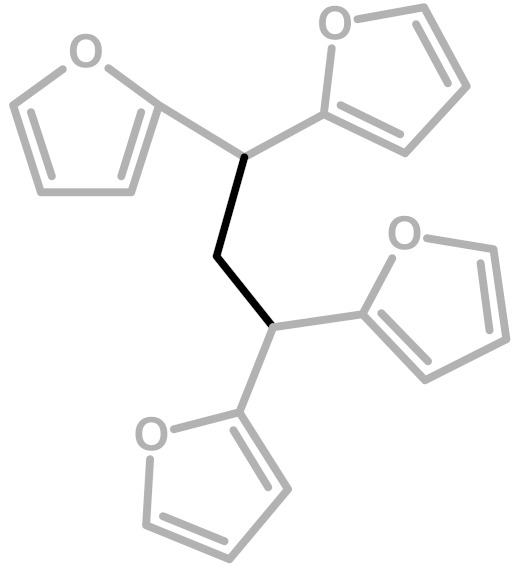	Chuang et al.—1984 [4]
5	Conjugated	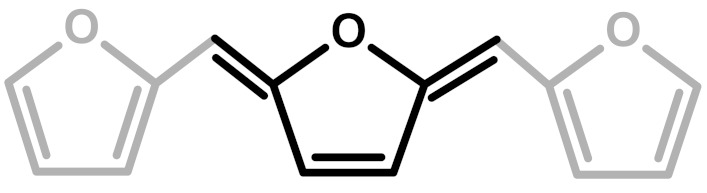	Buchwalter—1985 [5]
6	Diels-Alder	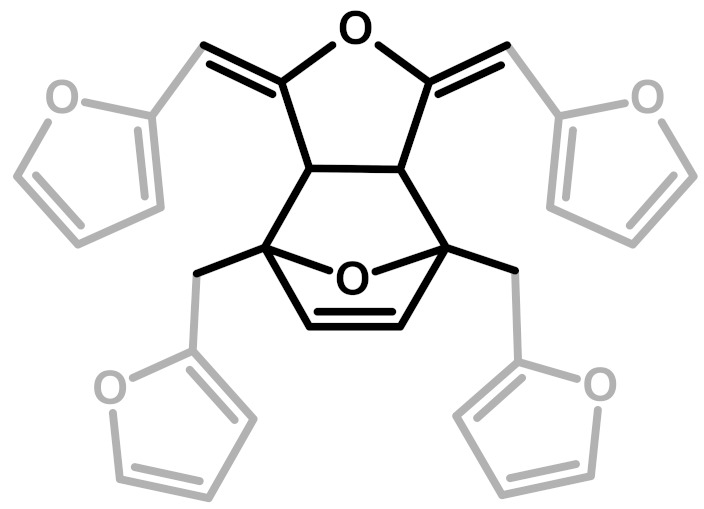	Choura et al.—1996 [6]
7	Ring-Opening + Diels-Alder	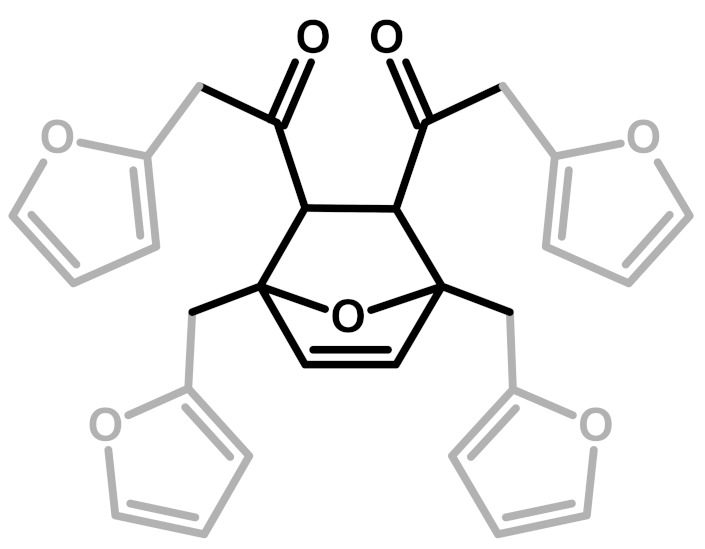	Present paper

**Table 2 polymers-11-02126-t002:** Summary of the attribution of the chemical shifts of PFA obtained by solid-state ^13^C-NMR.

Chem. Shift	Chemical Structures from Table 1							Attributions
	1	2	3	4	5	6	7	
219	No	**Yes**	No	No	No	No	**Yes**	C=O diketones vicinal dienophyle
204	No	**Yes**	No	No	No	No	**Yes**	C=O diketones
173	No	No	**Yes**	No	No	No	No	γ-lactones or Levulinic acid/ester
151	**Yes**	No	No	No	**Yes**	**Yes**	No	C2, C5 Furan
142	No	No	**Yes**	No	No	**Yes**	**Yes**	C=C in Diels-Alder bicycle or lactones
126	No	**Yes**	**Yes**	No	**Yes**	No	No	C=C in β between ketones or ester, C=C in the conjugated furan ring.
108	**Yes**	No	No	No	No	No	No	C3, C4 Furan
94	No	No	No	No	**Yes**	**Yes**	No	-CH= Bridge in conjugated systems
88	No	No	**Yes**	No	No	**Yes**	**Yes**	Quaternary C in Diels-Alder or lactones
49	No	**Yes**	No	No	No	No	**Yes**	-CH_2_- Bridge between furans & C=O
38	No	No	No	**Yes**	No	**Yes**	**Yes**	Tertiary C in Diels-Alder, -CH- bridge in methylene bridge (**4**)
27	**Yes**	No	**Yes**	**Yes**	No	**Yes**	**Yes**	-CH_2_- bridge in between furans, -CH_2_- bridge (**4**)
13	No	No	**Yes**	No	No	No	No	-CH_3_ in lactone structures

**Table 3 polymers-11-02126-t003:** Assignment summary of the hardened PFA ATR-FTIR absorption bands.

Wavenumber (cm^−1^)	Chemical Structures from Table 1							Attributions [11,30,31]
	1	2	3	4	5	6	7	
3150–3050	**Yes**	**Yes**	**Yes**	No	**Yes**	**Yes**	**Yes**	C-H stretching aromatic and vinyl
2950–2850	**Yes**	**Yes**	**Yes**	**Yes**	No	**Yes**	**Yes**	C-H stretching aliphatic
1790–1740	No	No	**Yes**	No	No	No	No	C=O stretching α,β-unsat γ -lactone
1720–1700	No	No	No	No	No	No	**Yes**	C=O stretching (isolated)
1690–1670	No	**Yes**	No	No	No	No	No	C=O vicinal to C=C (α,β-unsat. ketone)
1650	No	No	No	No	No	**Yes**	**Yes**	C=C stretching in D.A. (isolated) or conjugated diene
1615–1590	No	**Yes**	No	No	No	No	No	C=C vicinal to C=O (α,β-unsat. ketone) or conjugated diene
1560	**Yes**	No	No	No	**Yes**	No	No	C=C stretching (ring vibr. 2,5-disubstituted furans)
1510	**Yes**	No	No	No	**Yes**	No	No	C=C stretching (ring vibr. 2,5-disubstituted furans)
1450–1345	**Yes**	**Yes**	**Yes**	**Yes**	No	**Yes**	**Yes**	-CH_2_ scissoring and wagging
1345–1290	No	No	**Yes**	No	No	No	No	C-O stretching γ-lactone
1230–1100	No	No	No	**Yes**	No	**Yes**	**Yes**	Complex network of several vibrational modes associated with C-O ring stretching, C-C furan stretching,-CH_2_ in plane wagging.The peak at 1175 might be due to the C-O-C stretching of the D.A. (difficult to assign)
1100–1040	**Yes**	No	No	No	**Yes**	No	No	=C-O-C= ring vibration (associated with another peak in the range 1200–1120)
1012	**Yes**	No	No	No	No	No	No	-CH in plane wagging 2,5-disubstituted furan (Barsberg simulation)
980–900	No	**Yes**	No	No	No	No	**Yes**	-CH out of plane deformation vibration of alkenes –CH=CH– (usually 2 peaks and they are both present)
880–860	**Yes**	No	No	No	**Yes**	No	No	Furan ring C-H out-of-plane deform. vibration
810–745	**Yes**	No	No	No	**Yes**	No	No	Wagging/twisting -CH-ring structure
745–700	**Yes**	No	No	No	**Yes**	No	No	Furan ring -CH out of plane bend
677	No	**Yes**	**Yes**	No	No	**Yes**	**Yes**	-CH out of plane bending, cis -CH=CH-
599	**Yes**	No	No	No	**Yes**	No	No	Ring deformation vibration
550	No	Yes	**Yes**	No	No	**Yes**	**Yes**	-CH out of plane bending, cis -CH=CH-

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
