# Peer review of "Understanding the Polymerization of Polyfurfuryl Alcohol: Ring Opening and Diels-Alder Reactions"

_polymers, 2019, doi:10.3390/polym11122126_

Round 1

Reviewer 1 Report

The manuscript presented by Tondi et al. clearly gives new insights for persons working on the complex polymerisation of furfuryl alcohol. The complementary techniques (13C ss NMR, FT-IR, Raman) + the modelisation are very original.

Indeed the furan ring opening were expected for a long time (Conley et al.) but were only recently put forward to explain some behaviors (Falco et al.). This paper shows, from a structural point of view, that ring opening has clearly occured during polymerization and therefore cannot be neglected. It can be accepted for publication in Polymers

13 solid state NMR has the advantage of being "quantitative" so integration of signals (from spectra in Figure 1) can give additionnal information/discussions. For instance the carbonyl/furan ratio or the methyl/carbonyl ratio could be commented.

It could be interesting (in maybe another papers) to highlight to role played by the acidic initiator. Indeed sulfuric acid is not the most common catalyst employed for FA polymerization since it lead to fast polymerization which can somehow impact the structures.

Author Response

REV1

The manuscript presented by Tondi et al. clearly gives new insights for persons working on the complex polymerisation of furfuryl alcohol. The complementary techniques (13C ss NMR, FT-IR, Raman) + the modelisation are very original.

Indeed the furan ring opening were expected for a long time (Conley et al.) but were only recently put forward to explain some behaviors (Falco et al.). This paper shows, from a structural point of view, that ring opening has clearly occured during polymerization and therefore cannot be neglected. It can be accepted for publication in Polymers

Answer: Thank you very much!

13 solid state NMR has the advantage of being "quantitative" so integration of signals (from spectra in Figure 1) can give additionnal information/discussions. For instance the carbonyl/furan ratio or the methyl/carbonyl ratio could be commented.

Answer: Thank you for the suggestion. We thought about this possibility, but this approach would be too risky in our opinion. That’ s because different carbon surroundings affect the intensity of the signal in a way which is not univocally accepted. Usually quaternary carbons show less intense signal than primary ones, so as C=O vs. C- in the furan ring (especially by using cross-polarization set-up). Moreover, we do not know to which extent is this different, so we risk a hazardous estimation. In order to obtain really quantitative spectra it would afford prohibitively long relaxation, thus recording times. An alternative would be to add paramagnetic relaxation enhancing reagents, like Cr(acac)3, but this is not possible for solids.

It could be interesting (in maybe another papers) to highlight to role played by the acidic initiator. Indeed sulfuric acid is not the most common catalyst employed for FA polymerization since it lead to fast polymerization which can somehow impact the structures.

Answer: Thanks for the suggestion. The reviewer is perfectly right. Some study in this direction are already started and we wish to have some result soon.

Reviewer 2 Report

The authors combined 13C-NMR, FTIR, UV-Raman spectroscopies and computational modelling to try to give new insights in the complex polymerization mechanisms of furfuryl alcohol (FA) into Polyfurfuryl alcohol (PFA) in acidic conditions. The work is of scientific interest as it was shown that this polymerization is very complex and identification of final products is still under debate. The paper is very clear and well written. All the conclusions are supported by the data. The work indicate the possible formation of compound 7 (Ring-Opening + Diels-Alder) that was never mentionned before. The paper can be published in its present form.

Comment:

Despite quality and numerous data porvided in this work, some questions still remain. PFA was prepared by adding sulfuric acid to FA.The temperature of the reaction is not precised so I suppose that the reaction was done at room temperature ? Then, a post curing was performed at 103°C to complete the cure. As was shown by Guigo et al., the ratio between condensation reactions (leading mainly to the linear aliphatic polymer) and crosslinking strongly depends on curing temperature or curing cycles. Thus, we can suppose that different products will be obtained by varying the curing temperature. This point could be specified in the text, because some authors that will use a different temperature program could obtain different results. An other important factor that was not discussed in this paper, nor in previous one, is the influence of pressure (releases of H20), that certainly affect the reactions pathway.

Author Response

REV2

The authors combined 13C-NMR, FTIR, UV-Raman spectroscopies and computational modelling to try to give new insights in the complex polymerization mechanisms of furfuryl alcohol (FA) into Polyfurfuryl alcohol (PFA) in acidic conditions. The work is of scientific interest as it was shown that this polymerization is very complex and identification of final products is still under debate. The paper is very clear and well written. All the conclusions are supported by the data. The work indicate the possible formation of compound 7 (Ring-Opening + Diels-Alder) that was never mentionned before. The paper can be published in its present form.

Answer: Thank you very much!

Comment:

Despite quality and numerous data porvided in this work, some questions still remain. PFA was prepared by adding sulfuric acid to FA.The temperature of the reaction is not precised so I suppose that the reaction was done at room temperature ? Then, a post curing was performed at 103°C to complete the cure.

Answer: Yes, exactly. We specified  it in the text.

As was shown by Guigo et al., the ratio between condensation reactions (leading mainly to the linear aliphatic polymer) and crosslinking strongly depends on curing temperature or curing cycles. Thus, we can suppose that different products will be obtained by varying the curing temperature.

Answer: That’s correct. We believe that the temperature as well as nature and concentration of the acid affect the proportions between the different structures in the cured PFA. As already reported by Guigo ---

This point could be specified in the text, because some authors that will use a different temperature program could obtain different results. An other important factor that was not discussed in this paper, nor in previous one, is the influence of pressure (releases of H20), that certainly affect the reactions pathway.

Answer: This is also a nice point to be considered. We polymerized the FA in an open test-tube. But yes, I agree that we may expect different results in confined volumes.
